# COMPRESSED MAP PRIORS FOR 3D PERCEPTION

## ABSTRACT

Human drivers rarely travel where no person has gone before. After all, thousands of drivers use busy city roads every day, and only one can claim to be the first. The same holds for autonomous computer vision systems. The vast majority of the deployment area of an autonomous vision system will have been visited before. Yet, most autonomous vehicle vision systems act as if they are encountering each location for the first time. In this work, we present Compressed Map Priors (**CMP**), a simple but effective framework to learn spatial priors from historic traversals. The map priors use a binarized hashmap that requires only $32\text{KB}/\text{km}^2$, a $20\times$ reduction compared to the dense storage. Compressed Map Priors easily integrate into leading 3D perception systems at little to no extra computational costs, and lead to a significant and consistent improvement in 3D object detection on the nuScenes dataset across several architectures.

## 1 INTRODUCTION

Autonomous vehicles rarely visit a truly unseen location. Current deployments are typically geofenced to operate within a known, carefully mapped area. Later, fleet deployments will cover the same area over and over again, collecting massive amounts of rich sensor data from the same locations. Yet, current perception systems mostly treat the static world as never been seen before, and jointly infer both static and dynamic scene structures from sensor inputs alone Liu et al. (2022d); Peng et al. (2023); Wang et al. (2023a); Liu et al. (2022c); Huang & Huang (2022).

In this work, we introduce Compressed Map Priors (**CMP**) to equip vision models with a persistent memory of the world. We build up this memory via end-to-end training of the perception system using a rich and compact representation. To efficiently store and retrieve prior information, we employ a quantized spatial encoding that maps location queries to learnable features. We fuse these prior features with multi-view image features from existing perception architectures, allowing gradients to update the underlying embedding during training. Our map prior is trained end-to-end for the downstream task, allowing vision systems to utilize valuable information learned from past experiences to enhance its predictions.

At test time, the perception system leverages the learned map, enriched with a wealth of features built up during training. Through binary quantization of the learned embeddings, our prior map representation requires only 32KB/km² of coverage. This results in a $20\times$ reduction in memory compared to dense storage approaches, and introduces a $< 2\%$ computational overhead. This comprehensive prior knowledge serves to augment the capabilities of the underlying perception stack, allowing the system to make more informed and accurate inferences about the surrounding environment.

Our method is detector-independent and integrates seamlessly into existing perception systems with minimal architectural changes. To validate the efficacy of our approach, we con-

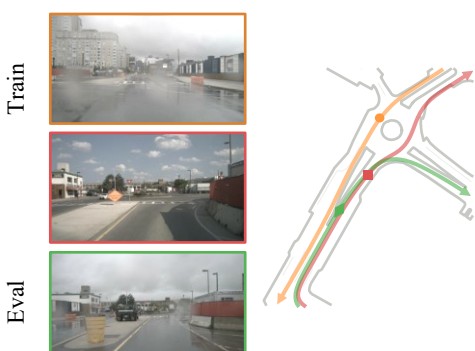

Figure 1: Multiple traversals of a scene in the nuScenes dataset Caesar et al. (2020) between the train and val split.

duct extensive experiments on the nuScenes Caesar et al. (2020) dataset and incorporate **CMP** into three distinct multi-view perception stacks. These baselines include both transformer-based Liu et al. (2022b) and convolutional Huang et al. (2021); Li et al. (2022c) perception systems. Our experiments show that incorporating **CMP** yields consistent performance gains across all evaluated baselines.

## 2 RELATED WORKS

**Camera-based 3D Perception.** Camera-only perception systems are a compelling choice for autonomous vehicles due to their high resolution and cost-effectiveness. While many highly accurate perception systems focused on monocular 3D object detection Wang et al. (2021a); Zhou et al. (2019); Park et al. (2021), modern autonomous vehicles utilize multiple cameras with potentially overlapping fields of view. To this extent, an increasing number of research efforts have shifted towards multi-view approaches Liu et al. (2022b); Li et al. (2022c); Xiong et al. (2023a); Huang et al. (2021); Yang et al. (2023c); Zhou & Krähenbühl (2022) which enable perception systems to construct a more comprehensive internal representation of the environment.

For multi-view perception, one line of work Philion & Fidler (2020); Li et al. (2022b); Reading et al. (2021) aggregates image features to a canonical "BEV" frame by predicting dense categorical depth for each image and pooling image features from a virtual frustum. Alternatively, BEV representations can be built using attention across camera views with geometric positional embeddings Xiong et al. (2023a); Chen et al. (2022a); Zhou & Krähenbühl (2022). Another approach Liu et al. (2022b); Wang et al. (2022b) bypasses the explicit BEV representation, directly forming object queries and attending to the multi-view images.

These models have been applied to a variety of tasks, showcasing their versatility and effectiveness in understanding the surrounding environment. Object detection Liu et al. (2022b); Li et al. (2022b); Wang et al. (2022b); Chen et al. (2022b) has been a primary focus, serving as a key role for autonomous vehicles. In addition, these models have been applied to HD-Map creation via semantic map prediction Philion & Fidler (2020); Li et al. (2022c); Zhou & Krähenbühl (2022); Liu et al. (2022d); Hu et al. (2021); Li et al. (2022a), and vectorized map prediction Li et al. (2022a); Liu et al. (2022a); Liao et al. (2022). These methods assume that each scene is encountered for the first time, overlooking valuable information from prior traversals. Our proposed **CMP** augments these models by integrating a persistent view of static scene elements from past traversals into the perception pipeline.

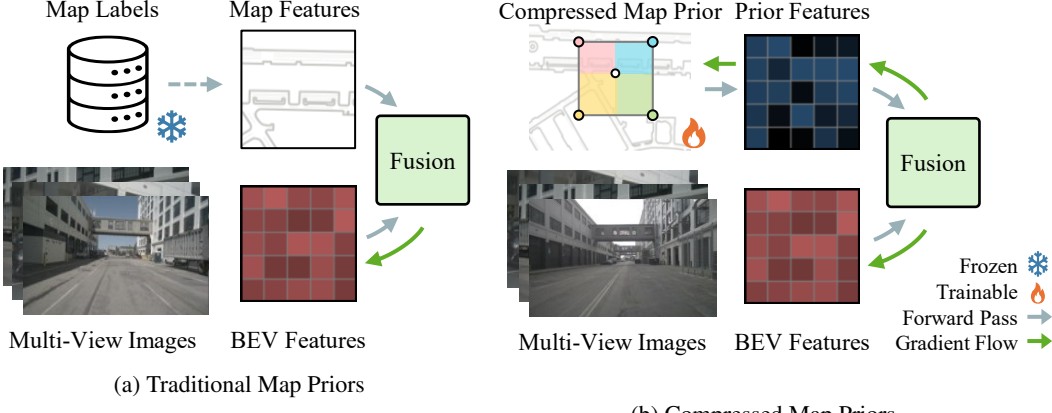

(a) Traditional Map Priors

(b) Compressed Map Priors

Figure 2: Comparison of how vision systems incorporate spatial priors. (a) Traditional approaches Yang et al. (2018) retrieve map annotations and fuse them with sensor features for downstream tasks. (b) Our proposed Compressed Map Priors (**CMP**) retrieves prior features by computing lookups from a highly compressed learnable embedding, represented by binarized features. **CMP** is fully differentiable and allows end-to-end learning with gradient updates through the entire pipeline.

**Perception with Historical Context.** Recent works Saha et al. (2021); Wang et al. (2023b); Yang et al. (2023a); Huang & Huang (2022) have developed models that ingest sensor input across multiple timesteps, demonstrating improvements over their single-frame counterparts. While these approaches focus on modeling temporal information, our work focuses on the *spatial* aspect, leveraging the fact that the same area is traversed multiple times.

Prior work has explored using maps as spatial priors for perception tasks. HDNET Yang et al. (2018) integrates HD map information into lidar features for 3D object detection. MapFusion Fang et al. (2021) extends this concept by predicting map information, removing the dependency on having maps available at inference time. Hindsight You et al. (2022) precomputes and stores quantized features from historical point cloud data. At test time, they augment the current scene with geo-indexed historical features, resulting in improved detection performance. This closely resembles our compressed map prior: Features computed at training time help inference at test time. The key difference is the representation of the prior. Hindsight uses an explicit point-based prior, while **CMP** uses a much more compact implicit function-based representation that is learned jointly with the perception system during training.

Recent works have also explored integrating historical data into camera-based perception systems. BEVMap Chang et al. (2024) rasterizes map annotations into each camera's perspective view to help reduce errors caused by depth ambiguity. NMP Xiong et al. (2023b) targets static semantic map segmentation from multi-view camera inputs by augmenting live sensor features with historical features using a GRU fusion module. They recursively update their prior by directly storing the features into the global map, represented as a set of dense "tiles", and show this external memory improves map segmentation performance. PreSight Yuan et al. (2024) utilizes historical data to model entire cities as a collection of high-fidelity neural fields Müller et al. (2022); Yang et al. (2023b). Their approach uses a two-stage process: first optimizing the representation with photometric reconstruction loss, then retrieving features from these learned representations to enhance online perception models.

**Scene Representation.** Neural scene representations have emerged as powerful tools for modeling 3D environments through learned implicit functions. NeRF Mildenhall et al. (2021) pioneered this approach by mapping 3D coordinates to learned features, producing density and color values for novel view synthesis. Recent advances have introduced sparse spatial structures Yu et al. (2022) and hash-based encodings Müller et al. (2022); Shin & Park (2023) that dramatically improve memory efficiency, enabling accurate reconstruction for larger scenes.

We draw a direct analogy between these representations and traditional maps: both serve as structured spatial priors providing contextual understanding of an environment. Just as neural implicit functions encode spatial features in a compressed and queryable format, maps encode structured priors that aid downstream reasoning. Building on this insight, our approach integrates compact spatial representations as the core mechanism for prior map representation. By encoding priors as implicit spatial embeddings, we achieve a scalable and efficient representation that balances expressivity and computational feasibility while remaining adaptable to large-scale environments.

## 3 PRELIMINARIES

**Multi-view 3D Object Detectors** use $n$ camera images $I_1, \ldots I_n$ with $I_k \in \mathbb{R}^{H \times W \times 3}$ and corresponding 2D pose information $p_k$ to predict 3D bounding boxes $B = \{b_1, b_2, \ldots, b_m\}$, where each bounding box $b_i$ is represented by its center, dimensions, orientation, and class score. Internally, the detector consists of two major components - a feature encoder $E$ and decoder head $D$. Within the encoder, an image backbone extracts multi-view image features $\mathbf{X}_i$. These multi-view features are pooled into a single intermediate representation $\mathbf{X}_{sensor}$ via a multi-view image to BEV transformationLi et al. (2022c); Zhou & Krähenbühl (2022). The decoder head $d$ decodes the intermediate representation into detection proposals. 3D detectors can be broadly categorized into two primary archetypes: dense grid-based detectors Huang et al. (2021); Li et al. (2022c); Yang et al. (2023a); Liu et al. (2022d) and query-based transformer detectors Wang et al. (2022b); Liu et al. (2022b;c).

Dense grid-based detectors represent the scene as a feature map $\mathbf{X}_{sensor} \in \mathbb{R}^{h \times w \times c}$, corresponding to a local map-region of size $h \times w$ around the vehicle. The decoder operates directly on this intermediate map representation and translates each spatial location into a potential detection with an associated score. Transformer-based detectors use a sparser intermediate representation in the

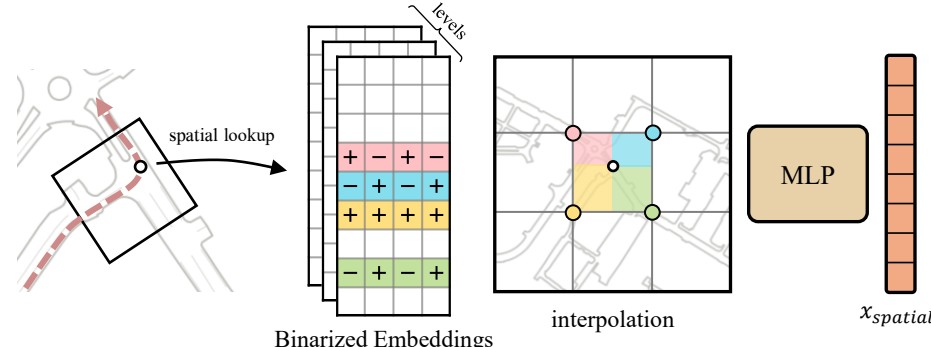

Figure 3: **Overview.** Illustration of our map representation. (1) We define a regular grid of query points in ego-vehicle coordinates. (2) For the four corners of cell this point falls in, we perform a spatial lookup in a binarized embedding. (3) Bilinear interpolation of the corner features produces a level-specific feature - we repeat this across multiple spatial resolutions and concatenate the features to produce a multi-scale feature representation. (4) Finally, we use an MLP to project the multi-scale feature into a single prior feature $x_{prior}$. We perform this in parallel for all query points to obtain the full prior $\mathbf{X}_{prior}$.

form of queries. They start from a set of $m$ learned queries $Q = \{q_1, \ldots, q_m\}$ which cross-attend to image features $\mathbf{X}_i$. The resulting sparse feature representation $\mathbf{X}_{sensor} \in \mathbb{R}^{m \times c}$ forms the input to a transformer-based decoder. BEVFormer Li et al. (2022c) uses concepts from both: a map-based BEV representation forms queries for a transformer-based multi-view encoder-decoder. Our compressed map prior applies to both kinds of architecture, albeit with a slightly different fusion architecture.

**Sparse representations for neural rendering.** In neural rendering, sparse representations Müller et al. (2022); Yu et al. (2022) have gained widespread adoption due to their memory efficiency. In particular, multi-resolution spatial hash encodings Müller et al. (2022) scale sublinearly with respect to the scene area. To encode a point $x \in \mathbb{R}^2$, it is first mapped onto four integer coordinates $y_1, \ldots, y_4 \in \mathbb{I}^2$. This is done by rounding $\frac{x}{\alpha}$ to the four corners of the surrounding hypercube. Here, $\alpha$ is the resolution of the spatial discretization. A spatial hash function $h : \mathbb{I}^2 \to \mathbb{I}$ then retrieves an embedding $\theta_{h(y_i)}$ for each coordinate. The embedding matrix $\theta \in \mathbb{R}^{T \times d}$ is learned. For a specific resolution $\alpha$ the final spatial hash embedding $m_\alpha(x, \theta)$ is a bilinear interpolation of the retrieved embeddings of each corner of the hypercube. This process is lossy, multiple integer coordinates may map to the same hash embedding. To compensate for the lossy nature of this hash, neural rendering approaches combine multiple hashes at different resolutions.

$$\{m_{\alpha_1}(x, \theta_1), \cdots, m_{\alpha_L}(x, \theta_L)\}.$$

Finally, a small MLP projects these concatenated features for downstream tasks like neural rendering. We denote the final multi-scale feature representation as $\mathbf{m}(x, \boldsymbol{\theta})$.

## 4 COMPRESSED MAP PRIORS

Our compressed map prior (**CMP**) consists of two components: A sparse and compressed map representation builds up a feature representation $\mathbf{X}_{prior}$ of the static scene. A light-weight fusion module then combines the spatial map features into an existing 3D detection architecture. See 2 for an overview. At training time, we differentiate through the map representation to learn a persistent prior that helps the 3D detector improve its accuracy.

### 4.1 PRIOR MAP REPRESENTATION

We retrieve prior features $X_{prior} \in \mathbb{R}^{h \times w \times 128}$ for a discretized $h \times w$ region $\mathbf{g} \in \mathbb{R}^{h \times w \times 2}$ centered around the ego-vehicle. Let $g_{ij}$ denote a single grid cell at location $i, j$. Using vehicle pose, we first convert $\mathbf{g}$ to global coordinates using the affine transform $M, t$ and retrieve the corresponding hash embedding $X_{prior} = \mathbf{m}(M\mathbf{g} + t, \boldsymbol{\theta})$.

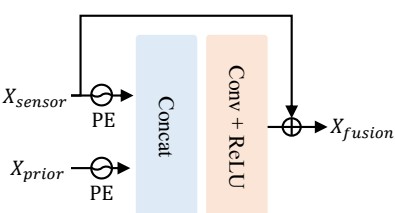
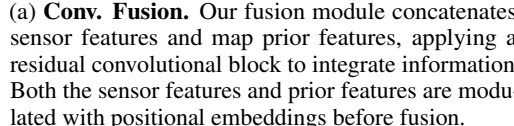
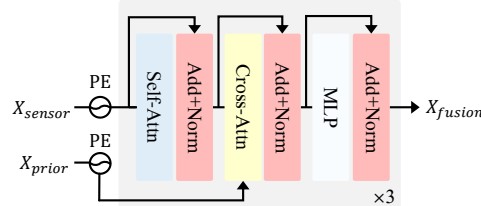

(a) **Conv. Fusion.** Our fusion module concatenates sensor features and map prior features, applying a residual convolutional block to integrate information. Both the sensor features and prior features are modulated with positional embeddings before fusion.

(b) **Token Fusion.** For transformer-based architectures, we use cross-attention to integrate map priors with sensor features. This allows sparse sensor embeddings to query relevant spatial context from our map representation.

To further compress the representation (critical for deployment constraints), we binarize features in the channel dimension: $\theta_k = \text{sign}(\tilde{\theta}_k)$. This discretization aligns naturally with the categorical nature of HD map annotations, which are inherently discrete semantic labels. During training, we materialize the underlying spatial embedding $\tilde{\theta}$ as real-valued parameters to allow for end-to-end learning via the straight-through estimator. During inference, we directly binarize these parameters, saving a substantial amount of memory compared to the full or half precision representations. For example, in the Boston area of the nuScenes Caesar et al. (2020) dataset, which spans 6.4 km², our approach requires only 32 KB/km² with binary quantization, compared to $\sim 1$ MB/km² without quantization and $\sim 125$ MB/km² for a dense representation at the same spatial resolution. This memory efficiency comes with minimal computational overhead; our efficient PyTorch implementation extracts the learned map prior with negligible computational overhead ($\sim 2\%$ of the total runtime.).

### 4.2 PRIOR FUSION

Modern detection architectures extract $\mathbf{X}_{sensor}$ from multi-view camera inputs and pass them to a downstream decoder. To integrate prior knowledge, we introduce a lightweight fusion module $F$ to incorporate the map prior features $\mathbf{X}_{prior}$ into the onboard sensor features $\mathbf{X}_{sensor}$.

For dense grid-based detectors Huang et al. (2021) and BEVFormer Li et al. (2022c), we align the size of the map representation $\mathbf{X}_{prior}$ to the size of the intermediate birds-eye-view sensor representation $\mathbf{X}_{sensor}$. We then concatenate the two and fuse them using a single $3 \times 3$ convolution followed by a ReLU nonlinearity. Empirically, we find that adding a positional embedding to both the map prior $\mathbf{X}_{prior}$ and sensor representation $\mathbf{X}_{sensor}$ improved the model's performance.

For transformer-based detectors Wang et al. (2022b); Liu et al. (2022b;c), we fuse the map-prior into the sparse sensor embeddings $\mathbf{X}_{sensor}$ for each query. We use a cross-attention layer to join the sensor and map prior features. Since the sparse sensor features $\mathbf{X}_{sensor}$ do not spatially align with our map prior $\mathbf{X}_{prior}$, we rely on a positional embedding for both sensor and prior features. For prior features, we learn the positional embedding as a set of free parameters $\mathbf{E}_{prior} \in \mathbb{R}^{x \times h \times d}$. For sensor embeddings, we use a linear projection of their corresponding positional embedding from the baseline detection decoder. Figure 4b shows an overview of the transformer-based fusion process.

To ensure the model remains robust in areas with limited prior coverage, we perform random patch masking on the prior features during training. We randomly mask patches in the prior features, replace the priors with a learned mask token. This simple augmentation allows the model to leverage priors when available, while still maintaining the performance of the baseline in novel environments.

## 5 EXPERIMENTS

We evaluate the efficacy of adding learned spatial priors with **CMP** for 3D object detection from multi-view camera images across a variety of architectures. We conduct experiments on the nuScenes Caesar et al. (2020) dataset, which consists of 850 scenes (700 training, 150 validation), covering two cities - Boston and Singapore. A large proportion of the scenes in the dataset have been traversed multiple times (see Appendix). Each scene is 20 seconds long, with 3D bounding box annotations at 2 Hz for

| Method | NDS ↑ | mAP ↑ | mAAE ↓ | mAOE ↓ | mASE ↓ | mATE ↓ | mAVE ↓ |
|---|---|---|---|---|---|---|---|
| CenterNet Zhou et al. (2019) | 0.328 | 0.306 | 0.716 | 0.264 | 0.609 | 1.426 | 0.658 |
| DETR3D Wang et al. (2022b) | 0.374 | 0.303 | 0.133 | 0.437 | 0.278 | 0.860 | 0.845 |
| FCOS3D Wang et al. (2021b) | 0.393 | 0.321 | 0.160 | 0.503 | 0.265 | 0.746 | 1.351 |
| PGD Wang et al. (2022a) | 0.409 | 0.335 | 0.172 | 0.423 | 0.263 | 0.732 | 1.285 |
| BEVDet Huang et al. (2021) | 0.383 | 0.302 | 0.190 | 0.600 | 0.282 | 0.724 | 0.852 |
| BEVDet + CMP | 0.423 | 0.323 | 0.193 | 0.520 | 0.289 | 0.627 | 0.757 |
| PETR Liu et al. (2022b) | 0.403 | 0.339 | 0.182 | 0.531 | 0.279 | 0.793 | 0.931 |
| PETR + CMP | 0.422 | 0.349 | 0.168 | 0.530 | 0.277 | 0.766 | 0.836 |
| BEVFormer Li et al. (2022c) | 0.425 | 0.329 | 0.149 | 0.428 | 0.275 | 0.754 | 0.789 |
| BEVFormer + CMP | **0.447** | **0.366** | 0.166 | 0.414 | 0.272 | 0.665 | 0.845 |

Table 1: **3D object detection results on nuScenes `val` set.** The primary evaluation metrics are in bold, underlined scores indicate best within each baseline comparison.

a total of 40 frames per scene. For each frame, we use the six multi-view camera images with their calibrated intrinsics and pose information and ego-pose.

**Metrics.** Aligning with the standard 3D detection evaluation methodology, we report mean Average Precision (**mAP**) across all 10 classes, calculated using ground plane center distance for matching predictions with ground truth labels. Additional metrics include five true positive metrics (ATE, ASE, AOE, AVE, AAE) for measuring errors in translation, scale, orientation, velocity, and attributes. The nuScenes Detection Score (**NDS**) Caesar et al. (2020) is a weighted sum of the mAP and the true positive metrics and provides a comprehensive evaluation of a model's performance.

### 5.1 IMPLEMENTATION DETAILS

We apply our method to three different 3D object detection architectures: BEVDet Huang et al. (2021), BEVFormer Li et al. (2022c), and PETR Liu et al. (2022b). These models represent the best-performing models for 3D object detection and cover the two most used architectural paradigms: dense BEV architectures and end-to-end transformer architectures. For each baseline, we use the single-frame model and use only multi-view camera inputs and pose information.

For our prior storage, as described in Section 4.1, we use a multi-level hash embedding with $L = 4$ levels each with $T = 2^{16}$ learned embedding of size 8 for a total of 32 dimensions. The finest resolution has a size of $1m^2$ per cell, and the coarsest resolution has a size of $25m^2$ per cell. The MLP consists of 3 layers and projects the retrieved embeddings to $\mathbf{X}_{prior} \in \mathbb{R}^{w \times h \times 128}$, where the width $w$ and height $h$ match the intermediate representation $\mathbf{X}_{sensor}$ of the baseline architecture. For BEVDet Huang et al. (2021), $w = h = 128$ while BEVFormer Li et al. (2022c) uses a slightly larger resolution $w = h = 150$. For PETR Liu et al. (2022b), we use a coarser resolution $w = h = 64$. Both the BEVDet and BEVFormer models use a ResNet-101 image backbone initialized with weights from a pre-trained FCOS3D Wang et al. (2021a), and PETR uses a VoVnet-99 Lee et al. (2019) initialized from a DD3D Park et al. (2021) checkpoint. We use standard BEV augmentations (flipping, scaling, rotating) and apply the same augmentation accordingly when retrieving the prior features. We refer the reader to the supplementary material for full implementation details.

Across all models, we train for 24 epochs using the AdamW Loshchilov & Hutter (2019) optimizer using a learning rate $2 \times 10^{-4}$ with a learning rate warmup followed by a cosine annealing schedule. All experiments are performed on a single-node machine with 8 Titan-V GPUs and a total batch size of 8. The full training duration is approximately 1 day.

### 5.2 QUANTITATIVE RESULTS

**Comparison with Baseline Detectors.** We compare the performance of adding **CMP** across BEVFormer Li et al. (2022c), BEVDet Huang et al. (2021) and PETR Liu et al. (2022b) for 3D object detection on the nuScenes Caesar et al. (2020) Dataset. Across all baselines, we observe consistent improvements in the main evaluation metrics (NDS and mAP). The two BEV-based architectures, BEVFormer and BEVDet, benefit most significantly from the addition of our proposed

| Method | NDS ↑ | mAP ↑ | KB/km$^2$ ↓ |
|---|---|---|---|
| BEVDet | 0.383 | 0.302 | - |
| BEVDet + GT Map | 0.409 | 0.316 | 732.4 |
| BEVDet + CMP | **0.423** | **0.323** | **31.6** |
| BEVFormer | 0.425 | 0.329 | - |
| BEVFormer + GT Map | 0.435 | 0.335 | 732.4 |
| BEVFormer + CMP | **0.447** | **0.366** | **31.6** |

Table 2: **Comparison with traditional map priors.** In the "GT Map" baseline, ground truth map labels are used as priors.

prior, with BEVDet showing the largest improvement (10.4% relative NDS and 7.0% relative mAP) and BEVFormer demonstrating substantial gains (5.2% relative NDS and 11.2% relative mAP). Architectures with explicit spatial BEV representations appear to benefit more from the prior as the prior features align effectively with the model's internal representation.

**Comparison with Traditional Map Priors.** We compare our approach against traditional categorical map priors (denoted as GT Map) by rasterizing map annotations for road dividers, lane dividers, pedestrian crossings, road segments, and lane markings. We encode the annotations with a $1 \times 1$ convolution and employ our fusion strategy to integrate their map features into the baseline detectors.

Along with detection metrics, we report how both prior representations' memory profile scales in KB/km$^2$. For traditional priors, each map cell stores a binary indicator $\{0,1\}^6$ for each of the 6 semantic classes and we compute the memory requirements as $log_2(6)$ bits / cell at a resolution of 1m$^2$. Results in Table 2 demonstrate that while conventional map priors yield modest performance gains over their respective baseline detectors, our learned priors deliver substantially higher relative improvements, while requiring 20× less memory overhead.

**Comparison with Prior Work.** We compare CMP to two works that employ priors for online perception. BEVMap Chang et al. (2024) incorporates prior information by rasterizing map annotations into both the perspective view of the cameras, and the intermediate BEV representation. NMP Xiong et al. (2023b) enhances online map perception by constructing spatial representations during inference, using location information to retrieve and aggregate features into a local BEV representation. However, their approach builds these representations entirely online during inference, discarding the spatial knowledge learned during training. Additionally, NMP detaches feature gradients during aggregation, training only the prior fusion module rather than enabling end-to-end optimization. To provide a fair comparison, we modify their method (denoted NMP*) to utilize the spatial representations learned during training, as detailed in the supplementary material.

We evaluate the methods across both 3D object detection and map segmentation. The map segmentation task consists of three classes: divider, pedestrian crossing, and road boundaries, and we report mean Intersection over Union (mIoU) across these classes as the primary metric. For this comparison, we use BEVFormer with a smaller spatial resolution of $w = h = 100$ as the backbone. We use the original detection head and add a segmentation head. We train the both ours and NMP jointly with the original detection loss and a weighted cross-entropy loss for segmentation.

We show the comparison between the different prior representations in Table 3. While all learned priors show various improvements over the baseline, our method achieves a more significant improvement in both detection and segmentation.

| Prior | NDS | mAP | mIoU |
|---|---|---|---|
| Baseline | 0.394 | 0.323 | 0.444 |
| BEVMap[†] | 0.398 | 0.319 | **0.711** |
| NMP* | 0.409 | 0.341 | 0.501 |
| CMP | **0.434** | **0.355** | 0.690 |

Table 3: **Comparison with learned priors.** [†] denotes using ground truth map labels as input.

| $T$ | Size (KB) | KB/km$^2$ | mIoU | Road | Divider |
|---|---|---|---|---|---|
| $2^{15}$ | 106.0 | 16.6 | 0.636 | 0.894 | 0.405 |
| $2^{16}$ | 202.0 | 31.6 | 0.670 | 0.909 | 0.476 |
| $2^{17}$ | 351.4 | 55.0 | 0.752 | 0.926 | 0.591 |
| $2^{18}$ | 607.4 | 95.0 | 0.785 | 0.930 | 0.662 |

Table 4: **Reconstruction experiments**

For the map prediction task, all the segmentation classes (divider, pedestrian crossing, road boundaries) are static, and our method can trivially learn to embed the map into the prior features. Moreover, using CMP shows a larger improvement in detection performance, demonstrating the effectiveness of our method as a prior for autonomous driving perception. We hypothesize that this improvement stems from our approach's ability to propagate gradients directly into the learned prior, unlike the baseline methods that rely on storing or retrieving the prior in a non-differentiable manner.

### 5.3 ABLATIONS

**Map Compression.** To evaluate the quality of our map representation and the effects of the spatial compression, we conduct a reconstruction experiment where we predict semantic map segmentation using *only* the learned prior features. We add a simple MLP probe to the prior features $\mathbf{X}_{prior}$ to predict the map segmentation and using a weighted cross entropy loss between the prediction and segmentation labels.

Table 4 shows our prior representation is able to faithfully reconstruct the global map, and has the capacity learn a latent representation of the map annotations. We observe that large-scale features like "road" classes achieve high IoU scores even at lower capacities, while fine-grained structures such as "divider" classes show more substantial improvements with increased representation size.

**Performance with Multiple Traversals.** We study how the number of traversals seen during training affects model performance, where a traversal is defined as each sample being less than 50m from any other sample seen during training (see supplementary for partitioning details). As shown in Figure 5, our method demonstrates the effectiveness of leveraging map priors in 3D perception.

**Performance vs. Distance.** We conduct an analysis across three distinct distance thresholds: "close" (0-10 meters), "medium" (10-25 meters), and "far" (25-50 meters). We measure the detection precision of the "car" class in Figure 6. In the "close" regime, both methods perform similarly, as vision provides sufficient signal for nearby objects. Notably, our method exhibits less performance degradation across distance ranges, indicating that the spatial prior provides valuable context for detecting distant objects where visual information is limited.

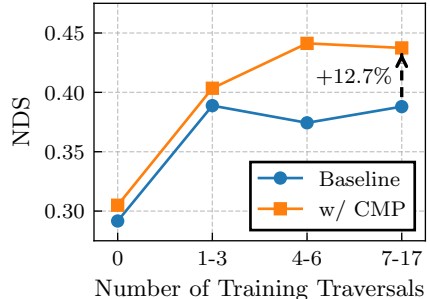

Figure 5: **Performance with different numbers of training traversals.** Both methods improve as traversals increase, but **CMP** significantly outperforms the baseline, with gains magnifying at higher traversal counts (0, 1-3, 4-6, 7-17 traversals). Our model gracefully degrades to baseline performance in areas with no prior traversals.

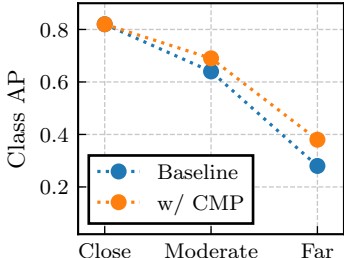

Figure 6: **Performance over different distances.** Average precision for the "car" class across three different distance thresholds: close (0-10 meters), medium (10-25 meters), and far (25-50 meters).

| Model | Binarized | NDS | mAP | Memory (KB/km$^2$) |
|---|---|---|---|---|
| BEVDet | ✓ | 0.426 | 0.323 | 32 |
| | | 0.431 | 0.322 | 640 |
| BEVFormer | ✓ | 0.447 | 0.366 | 32 |
| | | 0.450 | 0.365 | 640 |

Table 5: **Comparison of binarized vs. full-precision embeddings**

**Binary vs. Full Precision Embeddings.** We evaluate the impact of binarizing the spatial embeddings compared to using full-precision parameters. Table 5 shows that binarization maintains nearly identical performance while achieving substantial memory savings. The minimal performance degradation (0.8% NDS for BEVDet, 0.3% NDS for BEVFormer) demonstrates the effectiveness of our binary quantization strategy.

**Model Latency.** Table 6 shows the computation overhead of our method with respect to the baseline model's total latency. We provide timing results over 100 samples, measured on a single A5000 GPU using a batch size of 1. Incorporating our prior incurs a relatively small overhead, about $\sim 3\%$ of the full model's latency.

| Operation | Time (ms) | % Total |
|---|---|---|
| Prior Sampling | $7.63 \pm 0.09$ | 2.50% |
| Prior Fusion | $1.57 \pm 0.03$ | 0.51% |
| Full Forward | $305.17 \pm 0.37$ | - |

Table 6: **Computational Overhead**

## 6 CONCLUSION

We present a framework for incorporating historical context into perception models with Compressed Map Priors (**CMP**), which employs a multi-resolution hash-based spatial encoding with binary quantization to efficiently store and retrieve prior spatial information.

Experiments on the nuScenes dataset demonstrate that **CMP** generalizes across diverse architectures, delivering consistent improvements with simple modifications to the architecture. Our method is designed for environments where inference occurs in familiar, repeatedly traversed locations. While CMP requires additional training for changing/completely new environments, the modular design enables efficient adaptation: the majority of model parameters (backbone, fusion modules, detector head) can be frozen while optimizing only the prior parameters for the new locations.

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

## A ADDITIONAL ANALYSIS

### A.1 HASH EMBEDDING SIZE

The compressed map representation relies on a multi-level spatial hash encoding with hash table size $T$ as a key hyperparameter controlling the trade-off between representational capacity and memory efficiency. We systematically evaluate this trade-off by varying $T$ while keeping all other encoder hyperparameters fixed as specified in Section 5.1.

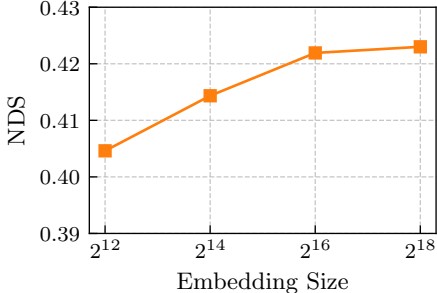

Figure 7: **Embedding Size.** The effect of embedding size $T$ vs. downstream detection performance.

Figure 7 demonstrates that increasing embedding size enhances the expressiveness of prior features, leading to improved detection accuracy. However, this improvement comes at the cost of increased memory requirements for storing the embeddings. Our experimental results reveal that an embedding size of $T = 2^{16}$ achieves an optimal balance between performance and memory efficiency, with diminishing returns observed for larger embedding sizes.

### A.2 COMPARISON WITH NEURAL MAP PRIORS

While our approach shares the high-level motivation of leveraging spatial priors with Neural Map Priors (NMP) Xiong et al. (2023b), several fundamental architectural and methodological differences distinguish our work and enable superior performance in the autonomous driving domain.

The most significant distinction lies in our training methodology. NMP employs gradient detaching during feature aggregation, training only the prior fusion module while keeping spatial features frozen. This design choice limits the model's ability to jointly optimize spatial representations with downstream task objectives. In contrast, our method enables end-to-end learning of positional embeddings alongside the detection task, allowing both prior parameters and fusion modules to be jointly optimized through the detection loss. This integrated optimization leads to more effective spatial representations tailored to the specific perception objectives.

Our approach also differs fundamentally in prior persistence and knowledge transfer. NMP constructs new priors during online inference, effectively discarding valuable spatial knowledge accumulated during training. This design is suboptimal for autonomous navigation scenarios where vehicles repeatedly traverse familiar routes. Our method maintains persistent priors that transfer learned spatial knowledge from training to inference, making it particularly well-suited for the predominantly repetitive traversal patterns in autonomous driving.

From a memory efficiency perspective, NMP relies on dense feature tiles that incur significant storage overhead. Our binarized hash-based encoding achieves a $20\times$ memory reduction (32 KB/km$^2$ vs 640 KB/km$^2$) while maintaining comparable detection performance. This efficiency gain is crucial for deployment in resource-constrained autonomous vehicle platforms.

## B EXPERIMENTAL DETAILS

### B.1 HYPERPARAMETER CONFIGURATION

We provide the experimental configuration used in our main results. Table 7 lists the key hyperparameters for our compressed map prior implementation and training procedure.

| Param | Value | Notes |
|---|---|---|
| $T$ | $2^{16}$ | Embedding size per level |
| $L$ | 4 | Hash resolution levels |
| $d$ | 8 | Embedding dimension |
| $\alpha_i$ | 1.0-25.0 | Hash level resolutions |
| $\eta$ | $2 \times 10^{-4}$ | Base learning rate |
| $\beta_{1,2}$ | $0.9, 0.999$ | Adam betas |
| $\lambda$ | 0.01 | Weight decay |
| $N_{\mathrm{mask}}$ | 0.25 | Prior masking ratio |
| $B$ | 8 | Batch size |
| $E$ | 24 | Training epochs |
| MLP | $[32, 32, 128]$ | Projection dimensions |
| $\mathbf{X}_{\mathrm{prior}}$ | $\mathbb{R}^{h \times w \times 128}$ | Prior dimensions |
| $\delta$ | 50m | Proximity threshold |

Table 7: Hyperparameter Configuration

## B.2 TRAVERSAL ANALYSIS METHODOLOGY

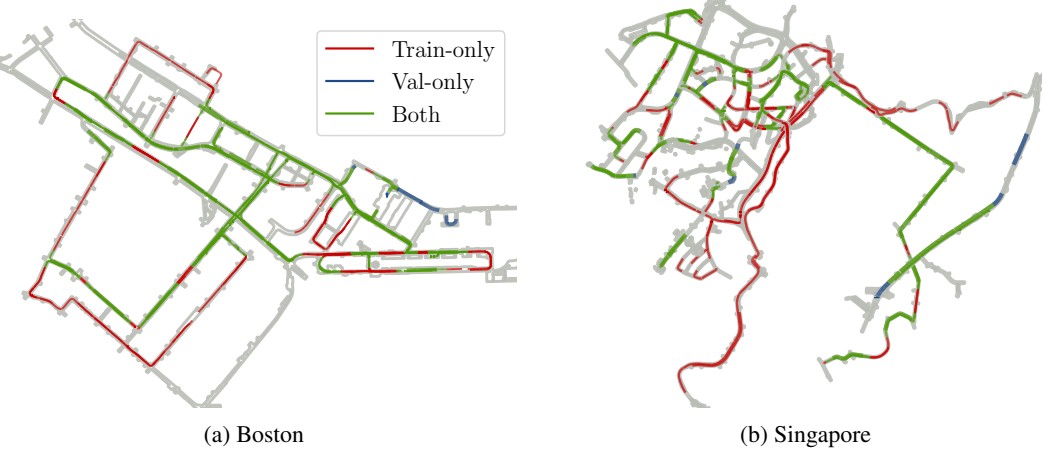

(a) Boston

(b) Singapore

Figure 8: **Map visualization of the nuScenes Caesar et al. (2020) dataset.** We delineate the traversals from the training and validation split of the dataset in bold colors. "Both" denotes scenes that have been traversed in both the training and validation splits. "Val-only" refers to scenes that have no significant overlap (within 50m) with any training scenes and are geographically disjoint from the training/validation set. "Train-only" refers to scenes that have no significant overlap with any validation scenes.

We define traversal frequency as the number of distinct training scenes that overlap with a specific validation sample location. To calculate this metric, we process the nuScenes dataset through the following steps:

First, we extract all scene samples with their corresponding timestamps $t_i$, position vectors $p_i \in \mathbb{R}^3$, and transformation matrices to the global coordinate frame from the nuScenes dataset. Due to data collection procedures, continuous trajectories are sometimes fragmented across multiple scene recordings. We merge related trajectory fragments using temporal proximity ($\Delta t < 10s$) and spatial proximity ($\|p_{i,end} - p_{j,start}\| < 10m$) thresholds to create contiguous scenes that better represent continuous traversals.

After the trajectories are merged, for each sample we count the number of unique training scenes containing at least one sample within a 50-meter radius. See fig. 9 for the full distribution.

Train Split

Figure 9: Distribution of traversal counts across the train split.

## C   USE OF LARGE LANGUAGE MODELS

In this work, LLMs were used for minor rephrasing and spell-checking assistance, specifically in the ablation analysis. We do not trust LLMs enough to use them in the results section.

