# OpenReview forum: "Compressed Map Priors for 3D Perception"
_ICLR.cc/2026/Conference — Submitted to ICLR 2026_

### Official Review · Reviewer_tZde · 2025-10-28

**Soundness:** 1
**Presentation:** 3
**Contribution:** 2
**Rating:** 2
**Confidence:** 4

**Summary:**

This paper proposes Compressed Map Priors, a framework that integrates spatial priors from historical traversals into multi-view 3D object detection for autonomous driving. CMP uses binary-quantized hash encoding to achieve extreme memory efficiency. This framework is compatible with both dense grid-based and transformer-based architectures. Experiments on nuScenes show consistent gains, outperforming traditional map priors and learned priors.

**Strengths:**

1. CMP’s binary hash embeddings achieve 32 KB/km² storage, 20× better than dense/GT Map, addressing on-board memory constraints. Its <2% runtime overhead also ensures real-time feasibility.
2. CMP adapts to both dense and transformer architectures with minimal modifications.

**Weaknesses:**

1. It is not a reasonable configuration to use learned prior map for 3D object detection. If a parked vehicle appears in both the training set and test set, it will be recorded in the prior map during training, making it easier to detect in the testing stage. However, this is meaningless for real-world autonomous driving, as such a vehicle may drive away at any time. Previous methods have not adopted a similar configuration. BEVMap utilizes the map annotations that do not contain object information to improve the accuracy of object detection. NMP leverages learned map priors, but these priors are used for the task of map segmentation.
2. The authors should provide more comparisons between CMP, NMP, and SOTA map segmentation methods in terms of map segmentation accuracy. The accuracy of "Val-only" and "Both" scenes can be presented separately.

**Questions:**

1. Why does the memory usage of GT maps much larger than that of CMP in Table 2? Is it possible to compress the GT map in the way of CMP?

---

> ### Author Response · Authors · 2025-11-25
> **Authors reply to R4 (tzde)**
>
> We thank the reviewer for their constructive feedback.
> We hope our detailed responses below address their concerns and clarify the contribution of our work.
>
> > W1: "It is not a reasonable configuration to use learned prior map for 3D object detection. If a parked vehicle appears in both the training set and test set, it will be recorded in the prior map during training, making it easier to detect in the testing stage. However, this is meaningless for real-world autonomous driving, as such a vehicle may drive away at any time. Previous methods have not adopted a similar configuration."
>
> We respectfully disagree with this characterization.
> Using map priors for 3D object detection is already an **established approach** in the literature (BEVMap, HDMapNet, Hindsight).
> Our contribution extends this paradigm to learned, memory-efficient spatial representations.
>
> That said, the reviewer raises a valid point about potential data leakage from overlapping parked vehicles within the experimental setup.
> To verify the performance gains are not due to memorization of parked vehicles, we performed a **detailed analysis on the distribution of parked vehicle instances** in the nuScenes dataset.
>
> **Parked vehicle analysis:**
> We first identified candidates for parked vehicles in the train and validation splits by filtering metadata of object instances by velocity and total displacement over time.
> This yielded 8,508 parked vehicles in train (1.6% of all vehicle detections) and 1,365 in validation (1.3% of all vehicle detections).
>
> For each validation parked vehicle, we then attempted to identify exact matches in the training distribution by computing bounding box IoU.
> 79 of the identified parked vehicles in the validation set (less than 0.1% of all validation vehicles) had significant spatial overlap (IoU > 0.5) with instances in the training set.
> We manually inspected the top 40 IoU matches by visualizing the camera sensor data and found that these instances were different vehicles parked in similar parking spots across different temporal captures, rather than memorization of specific vehicle instances.
>
> > W2: Additional comparisons on Map Segmentation
>
> We would like to point out that map segmentation comparisons would not effectively evaluate our contribution.
> Our reconstruction experiment (Table 4) demonstrates that priors can reconstruct map semantics using only localization information, without sensor data.
> We considered this in early development and chose to focus on detection (Table 1), where **spatial priors provide context but cannot directly solve the task**, requiring generalization to novel dynamic object configurations.
> We are happy to provide additional map segmentation analysis if the reviewer considers it critical.
>
> > Q1: GT Map Memory Usage + Is it possible to compress the GT map in a similar fashion?
>
> Ground truth map annotations are represented as dense raster tiles at fixed resolution, where each cell is stored independently.
> Our learned prior uses sparse hash embeddings that share features across locations and learn to resolve them through training.
> This spatial compression, along with our binarization, enables the large reduction in memory shown in Table 2.
>
> Regarding the reviewer's question about whether it is possible to compress GT maps similarly:
> Yes! Ground truth map annotations can be compressed in a similar fashion and the results are described in Table 4.

---

### Official Review · Reviewer_3XPW · 2025-10-30

**Soundness:** 2
**Presentation:** 3
**Contribution:** 2
**Rating:** 4
**Confidence:** 5

**Summary:**

The paper proposes Compressed Map Priors (CMP)—a multi-resolution spatial hash embedding that stores binarized per-cell features as a persistent prior, fused into standard camera-only 3D detectors. CMP is trained end-to-end (STE for binarization), adds negligible latency (~3%), and claims ~32 KB/km² storage (≈20× smaller than a dense alternative). On nuScenes, CMP consistently improves NDS/mAP across BEVDet, BEVFormer, PETR, and outperforms classical rasterized GT-map priors under a far smaller memory budget.

**Strengths:**

Pros:
1. Consistent accuracy gains on nuScenes across three diverse baselines; largest relative lift on BEV-style models.
2. Simple, detector-agnostic add-on: clean fusion blocks for BEV and transformer stacks (concat+Conv vs. cross-attention).
3. Thoughtful ablations: traversal count sensitivity and distance-band analysis support the “priors help when signal is weak/far”.

**Weaknesses:**

Cons:
1. While the method is shown across multiple camera-only 3D detectors, several stronger, recent baselines (e.g., StreamPETR, BEVNext) are missing. Without results on higher baselines, it’s hard to judge headroom and true practical impact.
2. The approach assumes AVs mostly drive in previously seen areas where priors exist; the conclusion also notes retraining/retuning is needed for new environments, which reduces universality.
3. No stress tests for prior dropout or corruption. In deployment, parts of the map prior can disappear or become stale. It’s unclear whether the system gracefully falls back to a normal 3D detector.
4. Train–val spatial leakage risk: Although Appendix details a 50 m overlap rule and a “val-only/both” partition, the main-paper results don’t report metrics conditioned on overlap. If CMP’s benefit concentrates where training traversals exist, the headline numbers could overstate generalization.
Ask to add: (a) per-split results: val-only vs. both; (b) performance vs. exact nearest-traversal distance (continuous, not bins).
5. Stale priors / change management :CMP encourages persistence. Lane closures, construction, snow, cones, parked trucks introduce map drift. Random patch masking helps, but it doesn’t simulate systematic stale bias.

**Questions:**

1. I am curious how you gate trust in the prior when sensor evidence contradicts it (e.g., temporary barriers). Is there a learned reliability head or confidence calibration for X_prior?
2. What is the failure mode when the car localizes off by one grid cell at the finest 1 m resolution? Any qualitative examples?
3. I am curious whether cross-attention fusion could over-attend to priors in sparse-query detectors. Did you observe reduced reliance when priors are masked at test time?
4. How much of the total memory footprint (per km²) includes the MLP projector and positional embeddings vs. the hash tables alone? Current reporting appears to count only the tables.
5. What is the performance about city-transfer: train priors in Boston only and evaluate on Singapore val-only segments; do CMP gains persist?
6. For fairness, BEVMap uses GT annotations (oracle). Why does CMP beat it on detection? Is it because CMP captures non-semantic cues (texture, curb geometry) that the 6-class raster misses? Could a richer GT (curbs, stop lines, sidewalks) close the gap?

---

> ### Author Response · Authors · 2025-11-25
> **Authors reply to R3 (3xpw)**
>
> We thank the reviewer for their thoughtful feedback.
>  We address their concerns below and believe these responses clarify our contributions.
>
> > W1: "Without results on higher baselines, it's hard to judge headroom"
>
> We appreciate the reviewer's feedback.
> During the rebuttal period, we started additional experiments to apply our method to the suggested StreamPETR and BEVNext codebases.
> We found that these codebases include several custom data/training modifications to the underlying training framework (`mmdet3d`, `mmcv`), requiring careful integration and validation to ensure correctness.
>
> We will continue these integrations through the rebuttal period, but respectfully note that these **baselines follow the architectural paradigms** already shown in our initial experiments.
>
> > W2: "Assumes AVs mostly drive in previously seen areas where priors exist"
>
> Please see the response to R1.W3 for the prior assumption.
> The challenge of adapting to changing environments is common to all systems using priors, including traditional HD maps and navigation systems.
>
> > W3: "Unclear whether the system gracefully falls back"
>
> In our initial submission, we show experiments for performance on areas with no traversals seen during training (Figure 5).
>
> To better understand whether the model gracefully falls back to baseline performance, we performed additional experiments and **fully masked the prior features during inference** by setting the mask ratio to `p = 1.0`.
> We found that the performance of the prior-less model almost exactly matches the baseline performance.
>
> | Method              | NDS   | mAP   |
> |---------------------|-------|-------|
> | Baseline            | 0.425 | 0.329 |
> | Fully Masked Prior  | 0.426 | 0.330 |
>
> > W4: "Trainval spatial leakage risk: Although Appendix details a 50 m overlap rule and a 'val-only/both' partition, the main-paper results don't report metrics conditioned on overlap. If CMP's benefit concentrates where training traversals exist, the headline numbers could overstate generalization."
>
> In our initial submission, we provided results conditioned on overlap in Figure 5.
>
> However, we may not have fully understood the reviewer's question and respectfully request that the reviewer clarify their concern.
>
> > Q1: How you gate trust in the prior when sensor evidence contradicts it
>
> While we do not explicitly gate trust, the fusion module *learns* to balance sensor and prior information through training.
> Random patch masking during training ensures the model maintains baseline performance when priors are corrupted or missing, effectively learning when to ignore them.
>
> > Q2: Performance with Localization Error
>
> Thank you for the suggestion!
> We ran an experiment to **evaluate performance under varying amounts of noise**.
> For each offset distance, we sample xy noise $v \sim \mathcal{N}(0, I)$ and normalize to $\|v\| = \text{offset}$ to simulate localization error.
>
> | Offset (m)    | NDS  | mAP  |
> |---------------|------|------|
> | 0.0           | 0.45 | 0.37 |
> | 0.5           | 0.44 | 0.36 |
> | 1.0           | 0.44 | 0.35 |
> | 2.5           | 0.43 | 0.33 |
> | 5.0           | 0.42 | 0.31 |
> | Baseline (N/A)| 0.43 | 0.33 |
>
> > Q3: Detailed Memory Footprint
>
> We use a small 2-layer MLP in the prior to project the features from `32 -> 128`, consisting of approximately 6K parameters.
> Trained with `bfloat16`, this corresponds to a fixed cost of `6000 * 2 / 1024 = 12 KB` per location.
> For the Boston location in nuScenes (6.4 square km), the hash embeddings alone use 202 KB, or ~31.5 KB / square km.
> Including the MLP parameters, the full prior uses ~33.4 KB / square km.

---

### Official Review · Reviewer_pzk6 · 2025-10-31

**Soundness:** 2
**Presentation:** 3
**Contribution:** 2
**Rating:** 4
**Confidence:** 3

**Summary:**

The paper introduces Compressed Map Priors (CMP), a data-driven approach for incorporating spatial priors into 3D perception models for autonomous driving. CMP leverages a multi-resolution hash-based embedding scheme with binary quantization to efficiently encode prior knowledge from repeated traversals of the same environment. The compressed prior is fused with standard 3D detection backbones and trained end-to-end with downstream detection losses. Extensive experiments on the nuScenes dataset show that CMP provides consistent detection gains, substantial memory savings (20x reduction), and minimal computational overhead, outperforming both traditional map priors and recent learned priors.

**Strengths:**

1. The idea of efficiently leveraging historical traversals to inform 3D perception systems addresses a fundamental inefficiency in current approaches, which often treat every scene as novel despite repeated exposure.
2. The architecture is modular and demonstrated to work across several leading baselines (BEVDet, BEVFormer, PETR), with minimal intrusion.
Thorough experiments: CMP is compared quantitatively against strong baselines, including modern learned and traditional map priors, across multiple architectures and uses appropriate metrics .

**Weaknesses:**

1. The method is explicitly described as being beneficial in well-traversed environments (Section 6), but its limitations in places with limited or no prior coverage are only superficially addressed via random patch masking.  No rigorous experiments or quantitative breakdowns for novel/unseen areas are provided, raising concerns for real-world deployment.
2. Though BEVFormer, PETR, and BEVDet are credible representatives, modern BEV occupancy grid predictors (such as OccFeat or PointBeV, referenced in the related web context) and object-centric methods (e.g., OC-SOP) are not included as comparative baselines in either main results or ablations, potentially missing stronger competitors or alternative design philosophies.

**Questions:**

1. Can the authors provide more rigorous or quantitative evaluations of CMP when applied to environments with entirely novel scenes (not traversed during training), explicitly reporting both absolute and relative performance drops?
2. Please clarify the impact of the random patch masking augmentation.  Have ablation studies been performed where this is turned off?

---

> ### Author Response · Authors · 2025-11-25
> **Authors reply to R2 (pzk6)**
>
> We thank the reviewer for the thoughtful review and address their concerns below:
>
> > W1: "No rigorous experiments or quantitative breakdowns for novel/unseen areas are provided, raising concerns for real-world deployment"
>
> We kindly refer the reviewer to our response to R1.W3 for the point regarding real-world deployment.
>
> For additional details on evaluations in novel/unseen areas, please see our response to R3.W3.
>
> > W2: Modern BEV occupancy grid predictors and object-centric methods are not included
>
> We thank the reviewer for the suggestion.
> These are important directions in AV perception.
> However, given the breadth of perception tasks (detection, tracking, forecasting, occupancy, segmentation), we focused our initial evaluation on 3D object detection to demonstrate the core contribution and leave additional tasks for future work.
>
> > Q1: Evaluations for entirely novel scenes
>
> We refer the reviewer to our response to R3.W3, where we provide additional experiments **evaluating performance on novel scenes**.
> We welcome any suggestions for additional experimental setups.
>
> > Q2: Significance of the Patch Masking Augmentation
>
> We thank the reviewer for the thoughtful question and are happy to clarify the significance of patch masking (Section 4.2).
>
> Our prior features are initialized randomly, and the model learns to update and fuse these features through the training process: the more frequently a given area appears in the training distribution, the more informative the features become (Figure 5).
> However, prior features in novel areas are essentially random noise and can potentially degrade the model's downstream predictions.
> Patch masking is a **simple augmentation to reduce the model's over-reliance on the prior**.

---

### Official Review · Reviewer_wb2r · 2025-10-31

**Soundness:** 2
**Presentation:** 2
**Contribution:** 2
**Rating:** 2
**Confidence:** 5

**Summary:**

This paper presents a framework for incorporating historical context into perception models with Compressed Map Priors, which employs a multi-resolution hash-based spatial encoding with binary quantization to efficiently store and retrieve prior spatial information.

**Strengths:**

1. The map prior encoding method proposed in this paper offers storage advantages compared to previous approaches.
2. The proposed method enables end-to-end optimization of map priors and perception tasks.

**Weaknesses:**

1. The proposed method was only tested on a single dataset and lacks validation on other mainstream datasets, such as KITTI, Waymo, Argoverse2, etc.
2. The experimental section compares against outdated methods and lacks comparisons with the latest state-of-the-art approaches.
3. The proposed method is limited to datasets where the training and testing data have overlapping areas on the map, making it difficult to apply in real-world open scenarios.
4. The proposed model contains numerous hyperparameters, and ablation studies are lacking for some of them.

**Questions:**

1. Please refer to the Weaknesses section.
2. Is the proposed method applicable to other 3D perception tasks such as 3D segmentation and tracking?

---

> ### Author Response · Authors · 2025-11-25
> **Authors reply to R1(wb2r)**
>
> We thank the reviewer for their time and constructive feedback.
>
> > W1: Single dataset evaluation
>
> We follow the standard evaluation practice of our baselines (BEVDet, PETR, NMP, BEVMap, BEVFormer), for which all but BEVFormer evaluate **exclusively on nuScenes**.
>
> > W2: Compares against outdated methods
>
> Our selected baselines (BEVDet, BEVFormer, PETR) cover the major architectural paradigms for camera-based BEV perception.
> The latest state-of-the-art methods still use the **same architectural principles**, and by evaluating on these core architectures, we can isolate the contribution of our spatial prior.
>
> > W3: Limited to datasets where the training and testing data have overlapping areas ... difficult to apply in real-world open scenarios
>
> As the reviewer notes, our proposed method requires repeated traversals between training and testing locations.
> However, we respectfully disagree that this is a major limitation.
> U.S. regulations require Level-4 AVs to operate within geofenced Operational Design Domains (ODDs), where deployments must **define and operate within clearly bounded geographic areas**.
> For example, California regulations require autonomous vehicle operators to define "boundaries of the ODD" and provide a "map of the Operational Design Domain" [1].
>
> [1] CPUC, "Application Guidance for AV Programs,"
> https://www.cpuc.ca.gov/-/media/cpuc-website/divisions/consumer-protection-and-enforcement-division/documents/tlab/av-programs/av-program-applications--general-guidance-august-2024.pdf
>
> > W4: Contains numerous hyperparameters, and ablation studies are lacking for some of them
>
> Our method's hyperparameters trade off compression ratio against representational power.
> In our initial submission, we provided ablations for the key design choices: compression ratio (Table 4, Appendix Figure 7) and the effects of binarization on the learned embeddings (Table 5).
> During the rebuttal period, we added ablations for the number of multi-resolution levels $L$ and kindly refer the reviewer to the updated appendix.
>
> We are happy to provide additional ablations for any specific hyperparameter that the reviewer has questions about!
>
> > Q1: Is the proposed method applicable to other 3D perception tasks?
>
> Yes.
> Modern architectures across 3D perception tasks (detection, tracking, forecasting, etc.) all operate on general intermediate "BEV" representations.
> Our proposed forms of fusion (Section 4.2) are designed to support the two major archetypes—query-based and dense architectures—and thus can be applied broadly to any task supported by the base architectures.

---

### Author Response · Authors · 2025-12-03
**Global Author Response**

We thank all reviewers for their time and constructive feedback.

**Discussion Period**

Due to the OpenReview security incident, the discussion period was cut short and the reviewers had limited time (~4 days) to provide feedback.
Unfortunately, we received no reviewer acknowledgement of our rebuttal prior to the freeze.

Overall, our proposed method offers a simple but effective approach to representing spatial priors.
Rather than relying on manually maintained HD maps (polygons and labels), we represent these priors as compact embeddings and learn them end-to-end.
We demonstrate that this approach consistently improves 3D object detection performance across all the backbones used in modern architectures.

**Key Clarification on Problem Setting (R1, R4)**

We believe the lower scores from R1 and R4 stem from their assessment that the problem setting is "not reasonable" or "difficult to apply in real-world scenarios".
We respectfully disagree.
AV deployments today **operate within fixed regions** where fleets repeatedly traverse the same mapped areas (see R1.W3), and our method is specifically designed around this real world setting.
Unfortunately, due to the discussion freeze, we were unable to discuss this directly with the reviewers.
We kindly ask the Area Chair to consider this in their evaluation.

**Additional Experiments and Ablations**

In our uploaded response, we have completed all requested experiments:
* **(R1) Hyperparameter Ablations:** We have ablation studies for the number of spatial resolutions in the hash embedding.
* **(R3) Full Prior Masking:** Added evaluation demonstrating that full prior masking recovers baseline performance exactly.
* **(R3) Sensitivity Analysis:** Presented a localization sensitivity evaluation.
* **(R4) Leakage Quantified:** Systematically analyzed train/val splits to quantify instance leakage, showing it affects **<0.1% of detections**.

We believe these additional experiments and clarifications address the core concerns from all reviewers.

Once again, we thank the reviewers and the Area Chair for their time in this unique* reviewing process.

---

### Meta-Review · Area_Chair_3s7k · 2025-12-26

**Summary:**

The reviewers raised four important concerns, including:
1) Motivation: This work assumes that autonomous vehicles only operate in areas with prior information, which limits the contribution.
2) Fundamental flaw: The method cannot handle failures resulting from changes in prior maps.
3) Advancement: No comparison with the latest state-of-the-art approaches.
4) Ablation: Lack of ablation analysis.

**Reviewer Concerns:**

The author's response fails to address any of these four concerns.

**Reviewer Scores:**

I think the reviewers will maintain their original ratings.

---

### Decision · Program_Chairs · 2026-01-26

Reject